# Iatrogenic Cerebral Amyloid Angiopathy After Childhood Brain Surgery: Novel Findings of MRI and CT

**DOI:** 10.3390/neurolint17050064

**Published:** 2025-04-24

**Authors:** Fumine Tanaka, Maki Umino, Megumi Matsukawa, Seiya Kishi, Ryota Kogue, Norikazu Kawada, Ken Kagawa, Takaya Utsunomiya, Hiroyuki Kajikawa, Hidehiro Ishikawa, Yuichiro Ii, Akihiro Shindo, Hajime Sakuma, Masayuki Maeda

**Affiliations:** 1Department of Radiology, Mie University School of Medicine, 2-174 Edobashi, Tsu 514-8507, Mie, Japan; m-tochio@clin.medic.mie-u.ac.jp (M.U.); megumim@med.mie-u.ac.jp (M.M.); r-kogue@med.mie-u.ac.jp (R.K.); sakuma@med.mie-u.ac.jp (H.S.); 2Department of Radiology, Matsusaka Chuo General Hospital, Matsusaka 515-0818, Mie, Japan; skishi@med.mie-u.ac.jp; 3Department of Neurology, Matsusaka Chuo General Hospital, Matsusaka 515-0818, Mie, Japan; n-kawada@mch.miekosei.or.jp; 4Department of Neurology, Suzuka Kaisei Hospital, Suzuka 513-0836, Mie, Japan; domtrek@msn.com; 5Department of Neurology, Mie University School of Medicine, Tsu 514-8507, Mie, Japan; utsunomiya-t@med.mie-u.ac.jp (T.U.); h-kajikawa@med.mie-u.ac.jp (H.K.); hidehiro-i@med.mie-u.ac.jp (H.I.); a-shindo@med.mie-u.ac.jp (A.S.); 6Department of Neuroimaging and Pathophysiology, Mie University School of Medicine, Tsu 514-8507, Mie, Japan; ii-y@med.mie-u.ac.jp; 7Department of Neuroradiology, Mie University School of Medicine, Tsu 514-8507, Mie, Japan; mmaeda@med.mie-u.ac.jp

**Keywords:** iatrogenic cerebral amyloid angiopathy, magnetic resonance imaging, computed tomography

## Abstract

**Background/Objectives**: A subtype of cerebral amyloid angiopathy (CAA), iatrogenic CAA (iCAA), has been increasingly reported. iCAA occurs primarily in patients who underwent surgery during childhood and is caused by the prion-like propagation of amyloid beta. This subtype of CAA tends to develop at a younger age than age-related CAA, usually before the age of 55. After a latency period of 20–40 years following surgery, it manifests as lobar intracerebral hemorrhage (ICH), cognitive impairment, or transient focal neurological episodes. Between 2023 and 2024, we observed four cases of possible iCAA, all of which had a history of neurosurgery during childhood. **Case presentation**: MRI findings for all cases revealed multiple lobar microbleeds. Two cases also showed cortical superficial siderosis and lobar ICH. Notably, contrast-enhanced 3D FLAIR demonstrated sulcal enhancement in two cases, and CT demonstrated cortical calcification in the bilateral posterior lobes in one case. **Conclusions**: Sulcal enhancement on contrast-enhanced 3D FLAIR and cortical calcification in the bilateral posterior lobes on CT may suggest advanced CAA in the present cases.

## 1. Introduction

Cerebral amyloid angiopathy (CAA) typically develops as a result of pathological changes in leptomeningeal and small cortical vessels, leading to intracerebral hemorrhage (ICH) and cognitive decline in elderly patients. Recently, a subtype of CAA, iatrogenic CAA (iCAA), has been reported worldwide [1]. iCAA is presumably caused by a prion-like transmission form of amyloid beta (Aβ) and occurs primarily in patients who underwent neurosurgery during childhood [1]. In clinical practice, iCAA may present as ICH, cognitive impairment, or transient focal neurological episodes (TFNEs) [1]. The onset of iCAA tends to occur before the age of 55 but can occur at the age 65 or older, depending on the timing of surgery and a latency period of 20–40 years [1,2]. iCAA has been frequently reported to occur with cadaveric materials such as Lyodura [2,3]. In addition, other reported causes of iCAA include the use of pituitary-derived hormones and blood transfusions [1]. iCAA can also occur in patients who have previously undergone neurosurgery, even without exposure to cadaveric materials [2,3].

The diagnostic criteria for iCAA have been proposed [1], and they include imaging features such as lobar ICH, lobar cerebral microbleeds (CMBs), and cortical superficial siderosis (cSS), which are the same as those used for the diagnosis of sporadic CAA in Boston criteria 2.0, except for non-hemorrhagic lesions [4]. As CAA is considered a type of small vessel disease (SVD), previous literature proposed the CAA-SVD scoring system, which consists of lobar CMBs, cSS, perivascular spaces in the centrum semiovale (CSO-PVSs), and white matter hyperintensity (WMH) [5]. This score ranges from 0 to 6, with higher scores indicating greater severity of SVD [5], which is useful for sporadic CAA.

Here, we report four cases of possible iCAA, which showed common imaging features of iCAA, and novel imaging findings on contrast-enhanced 3D fluid-attenuated inversion recovery (FLAIR) and CT that may contribute to a better understanding of iCAA.

## 2. Case Report

### 2.1. Case 1

A 47-year-old woman presented with severe cognitive impairment, including episodes of not knowing where her house was, indicative of loss of orientation. Her Mini-Mental State Examination score was 14 out of 30. She underwent surgery for cerebellar medulloblastoma at age five, followed by whole-brain radiation therapy and shunting for obstructive hydrocephalus. Cadaveric dura mater use was not confirmed. She had no history of hypertension or familial CAA. Susceptibility-weighted imaging (SWI) showed multiple cSS in both cerebral hemispheres as well as multiple CMBs located mainly in the cortex, and several microbleeds were detected in the cerebellum (Figure 1). The CAA- SVD score was 5 based on the previous literature [5], consisting of two points for lobar CMBs, two points for cSS, zero points for CSO-PVSs, and one point for WMH. Contrast-enhanced 3D FLAIR showed sulcal enhancement extending from the temporal to occipital lobes bilaterally while enhancement was not detected on the contrast-enhanced 3D gradient echo (GRE) T1-weighted image. Additionally, the T1-weighted image shows global brain atrophy. Neither cerebrospinal fluid (CSF) examination nor amyloid positron emission tomography (PET) was performed. Thus, this case was considered a possible iCAA based on the criteria [1].

### 2.2. Case 2

A 38-year-old man was found collapsed on the stairs and subsequently admitted to our hospital. On admission, right upper and lower extremity paresis was noted. He underwent left parietal craniotomy for a traumatic head injury at the age of six months. Cadaveric dura mater use was not confirmed. He had no history of hypertension or familial CAA. CT showed a lobar ICH in the left parietal lobe, and SWI showed multiple cortical CMBs adjacent to the ICH (Figure 2). The CAA-SVD score was 2 based on the previous literature [5], consisting of two points for lobar CMBs, zero points for cSS, zero points for CSO-PVSs, and zero points for WMH. Neither CSF examination nor amyloid PET was performed. Thus, this case was considered a possible iCAA based on the criteria [1].

### 2.3. Case 3

A 47-year-old man presented with dysarthria and transient symptoms of weakness and numbness in the left upper extremity. He had a history of a cerebellar tumor that had been removed at the age of 11 years. Use of cadaveric dura mater, Lyodura, was confirmed, although any other detailed information about surgical procedure was not available. Radiation therapy was not performed. He had no history of hypertension or familial CAA. T2*-weighted images showed no cSS in the right precentral gyrus or other gyri, while multiple lobar CMBs were detected mainly in both cerebral hemispheres, with several CMBs also observed in the cerebellum (Figure 3). The CAA-SVD score was 2 based on the previous literature [5], consisting of two points for lobar CMBs, zero points for cSS, zero points for CSO-PVSs, and zero points for WMH. CT showed prominent cortical calcifications in both occipital lobes, which were not visible on the T2*-weighted image. Neither CSF examination nor amyloid PET was performed. Thus, this case was considered a possible iCAA based on the criteria [1].

### 2.4. Case 4

A 56-year-old man presented with sensory and motor disturbances in his left hand, which subsequently improved without specific treatment. At the age of four, he suffered a traumatic brain injury in a car accident that required cranial surgery at the right parietal region. Cadaveric dura mater use was not confirmed. He had no history of hypertension or familial CAA. CT showed lobar ICH in the right cerebral hemisphere, and SWI showed multiple lobar CMBs and cSS in both cerebral hemispheres (Figure 4). The CAA-SVD score was 6 based on the previous literature [5], consisting of two points for lobar CMBs, two points for cSS, one point for CSO-PVSs, and one point for WMH. Notably, cSS was also observed in the right postcentral gyrus, which may be associated with TFNEs. Contrast-enhanced 3D FLAIR showed enhancement in the sulci of the bilateral occipital lobes while enhancement was not detected on the contrast-enhanced 3D GRE T1-weighted image. Neither CSF examination nor amyloid PET was performed. Thus, this case was considered a possible iCAA based on the criteria [1].

The clinical features and main imaging findings of all cases are summarized in Table 1.

## 3. Discussion

We reported four cases diagnosed with possible iCAA following neurosurgery during childhood. Although there have been many reports on iCAA, there have been no reports on MR findings such as those in the present case, in which contrast enhancement of the hemispheric sulci was observed on contrast-enhanced 3D FLAIR and cortical calcification was observed on CT.

Although only one study has reported about sulcal contrast enhancement in patients with iCAA [6], a previous study has documented this finding in cases of CAA-related inflammation [7]. One proposed hypothesis is that this finding may be due to increased meningeal vascular permeability due to disruption of the blood–meningeal barrier, leading to local leakage of gadolinium-based contrast agents into the subarachnoid space [8]. Cases 1 and 4 in this report have higher total CAA-SVD scores, suggesting severe CAA-related microangiopathy [9]. Thus, sulcal enhancement on contrast-enhanced 3D FLAIR may reflect vulnerable blood vessels severely affected by CAA.

CAA-related inflammation is reportedly part of the inflammatory spectrum associated with Aβ, occasionally referred to as spontaneous amyloid-related imaging abnormalities (ARIA)-like presentations [7]. A previous study demonstrated that patients with CAA-related inflammation may show contrast enhancement on T1 post-gadolinium sequences, suggestive of leptomeningeal involvement, which closely resembles our finding in iCAA [7]. Both iCAA- and CAA-related inflammation may share a common pathophysiological mechanism involving the clearance Aβ peptide [6]. Thus, differentiating iCAA- from CAA-related inflammation, or vasculitis, such as Aβ-related angiitis or primary central nervous system vasculitis, may be challenging based solely on the finding of leptomeningeal enhancement on MRI. Importantly, CAA-related inflammation and vasculitis typically present with WMH that is confluent or in large distribution throughout the cerebral hemispheres, and they tend to respond well to treatment, such as immunosuppressive or steroid therapy, which can be useful in distinguishing them from iCAA [7]. Additionally, lobar hemorrhage is more frequent in CAA than in CAA-related inflammation and vasculitis [10]. Notably, in our cases 1 and 4, lobar CMBs were observed near the leptomeningeal enhancement. In the previous study on iCAA, patchy subcortical T2/FLAIR hyperintensity or cortical edema was observed at the same site as leptomeningeal contrast enhancement, suggesting inflammation [6]. In contrast, our cases 1 and 4 have no T2/FLAIR hyperintensity or cortical edema indicative of inflammation. This difference may be because in the present case, sulcal enhancement was depicted using contrast-enhanced 3D FLAIR, which is highly sensitive to contrast leakage into the CSF, whereas 3D GRE T1-weighted image did not reveal sulcal enhancement. Thus, leptomeningeal enhancement in patients with iCAA does not necessarily indicate the inflammatory state although histopathological validation is unknown.

Another notable radiological finding in this report is the cortical calcification in the bilateral occipital lobes shown on CT in case 3, and this is considered to be the first report about this finding in iCAA. The cortical calcification is considered to be infiltrating vascular calcification, suggesting an advanced stage of CAA [11]. This finding has been reported in hereditary CAA, such as the Dutch type, but is also rarely reported in sporadic CAA, possibly due to the more aggressive nature and earlier onset of hereditary CAA [11,12,13,14,15]. iCAA has prolonged exposure to Aβ compared to sporadic CAA, which is similar to hereditary CAA. However, the total CAA-SVD score in case 3 was only 2, which may not indicate an aggressive course in terms of pathophysiology. Thus, further pathophysiological investigations are required to elucidate the relationship between the occipital calcification and the traditional findings of SVD. If the above novel findings are detected in patients with iCAA, careful follow-up is required, as lobar ICH tends to recur more frequently than sporadic CAA [2].

Common radiological findings of iCAA are similar to those of age-dependent sporadic CAA, such as CMBs and cSS, except for changes due to surgery [1]. Cases 1 and 3 showed several cerebellar CMBs, which are presumably not related to surgical procedures. Cerebellar cSS is reportedly a sign for advanced iCAA [16]. CMBs and cSS in iCAA tend to occur near the site of surgery, primarily extending to both cerebral hemispheres and occasionally to the cerebellum. Importantly, when CMBs and superficial siderosis are observed in a case with a history of radiation therapy, trauma, or surgery, radiation-induced vasculopathy and traumatic changes should be considered the main differential diagnoses. CMBs due to radiation-induced vasculopathy can occur anywhere within the irradiated area [17]. Regarding traumatic microbleeds, including those caused by diffuse vascular injury, microbleeds are typically punctate or linear in shape and are located in the cortical gray matter to white matter [18,19]. Superficial siderosis can be detected in cases after traumatic subarachnoid hemorrhage or surgery. However, the location of hemosiderin deposition on the surface of the brain is reportedly correlated with the initial bleeding sites and volume [20]. Thus, cSS detected not only in the ipsilateral hemisphere, but also in the contralateral hemisphere, is a key finding suggestive of CAA, as seen in case 1 and case 2 in this report. Furthermore, cSS in CAA can cause frequent TFNEs while superficial siderosis due to traumatic or surgical bleeding does not cause TFNEs, which is considered useful for diagnosis [21]. However, it should be noted that its symptoms of traumatic bleeding may occasionally resemble those of TFNEs [21]. Among the cases in this report, case 1 had prominent global brain atrophy, which may contribute to the major symptom of cognitive impairment. Cortical atrophy is one of the magnetic resonance imaging findings of iCAA, as well as typical sporadic CAA [22].

There are several limitations in our cases. All cases have no confirmation of amyloid by brain biopsy, amyloid PET, or CSF examination, and no genetic testing was performed. Not all cases have the contrast-enhanced 3D FLAIR and CT, and two cases had no calcification on CT. Further pathological investigations are required to elucidate the entire mechanism of sulcal enhancement on contrast-enhanced 3D FLAIR and cortical calcification in the bilateral occipital lobes on CT.

## 4. Conclusions

Since iCAA may be one of the important causes of lobar ICH and cognitive impairment in younger generations compared with sporadic CAA, there is a need to understand imaging markers of advanced-stage CAA using contrast-enhanced 3D FLAIR and CT. Additionally, the fluid biomarkers and pathological data are needed to confirm the iCAA hypothesis and diagnosis.

## Figures and Tables

**Figure 1 neurolint-17-00064-f001:**
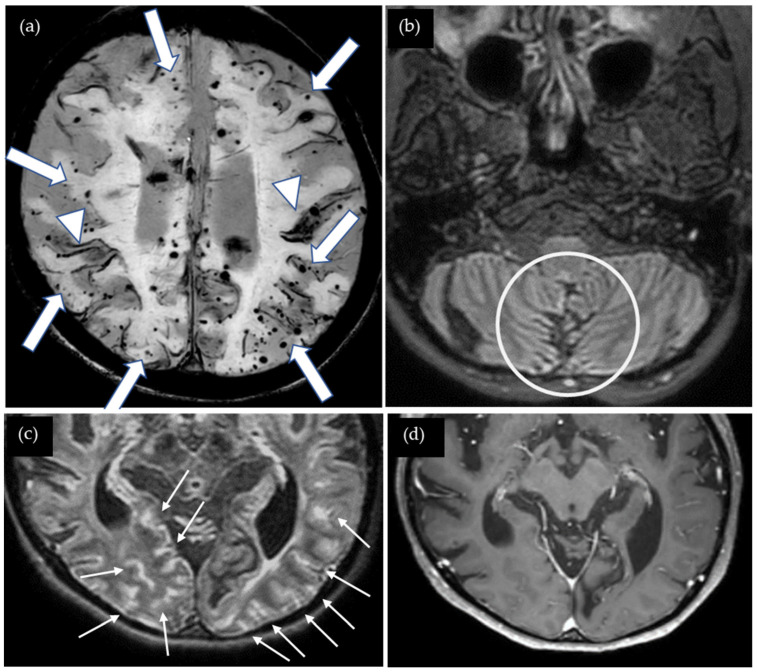
(**a**) SWI shows numerous lobar cerebral microbleeds (CMBs) (arrows) and cortical superficial siderosis (cSS) (arrow heads); (**b**) non-contrast 3D FLAIR shows postoperative changes in the cerebellum (circle); (**c**) contrast-enhanced 3D FLAIR shows sulcal enhancement in the bilateral occipital lobes (small arrows) while enhancement was not detected on (**d**) the contrast-enhanced 3D gradient echo (GRE) T1-weighted image.

**Figure 2 neurolint-17-00064-f002:**
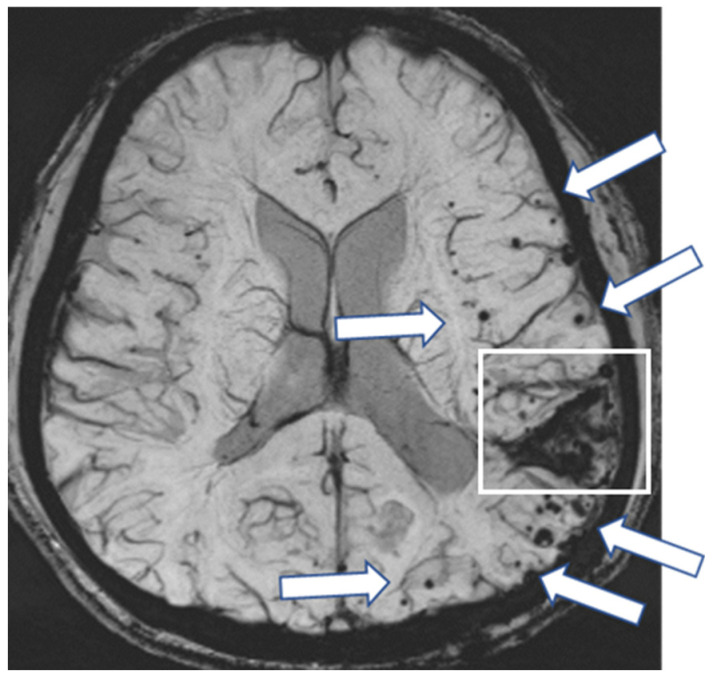
SWI shows multiple CMBs (arrows) adjacent to the lobar hemorrhage in the left parietal lobe (square).

**Figure 3 neurolint-17-00064-f003:**
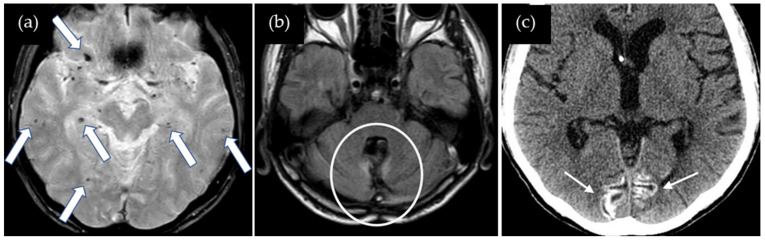
(**a**) T2*-weighted image shows no cSS in the right precentral gyrus or other gyri, while multiple lobar CMBs (arrows) are detected in both cerebral hemispheres; (**b**) 2D FLAIR shows postoperative changes in the cerebellum (circle); (**c**) CT shows prominent cortical calcifications in both occipital lobes (small arrows).

**Figure 4 neurolint-17-00064-f004:**
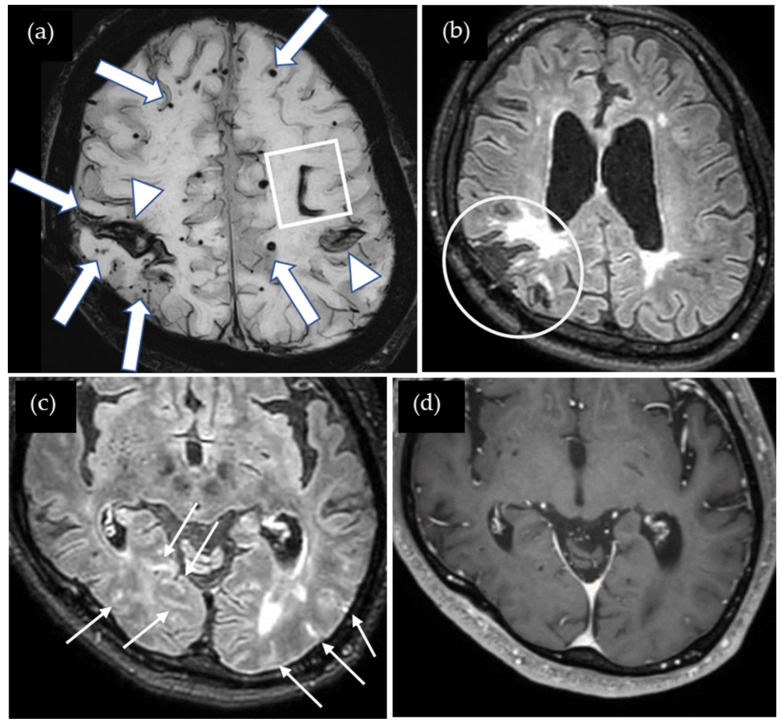
(**a**) SWI shows lobar hemorrhages (square) in both cerebral hemispheres, multiple lobar CMBs (arrows), and cortical superficial siderosis (cSS) (arrow heads). cSS in the right postcentral gyrus may be associated with TFNEs; (**b**) non-contrast 3D FLAIR shows postoperative changes in the right parietal lobe (circle); (**c**) contrast-enhanced 3D FLAIR shows sulcal enhancement in the bilateral occipital lobes (small arrows) while enhancement was not detected on (**d**) the contrast-enhanced 3D GRE T1-weighted image.

**Table 1 neurolint-17-00064-t001:** Summary of clinical features and imaging findings.

	Case 1	Case 2	Case 3	Case 4
Age at first presentation, years old	47	38	47	56
Sex	Female	Male	Male	Male
Age at exposure	5 years old	6 months old	11 years old	4 years old
Latency, years	42	38	36	52
Type of surgery	Resection of cerebellar medulloblastoma	Left parietal craniotomy due to a traumatic injury	Resection of cerebellar tumor	Right parietal craniotomy due to a car accident
Confirmed cadaveric dura mater use	unknown	unknown	used (Lyodura)	unknown
Presenting symptom				
Cognitive impairment	+	−	−	−
TFNE/transient symptoms	−	−	+	+
Extremity paresis	−	+	−	−
MRI findings				
Lobar ICH	−	+	−	+
Lobar CMB points	2	2	2	2
cSS points	2	0	0	2
CSO-PVS points	0	0	0	1
WMH points	1	0	0	1
CAA-SVD score	5	2	2	6
Sulcal contrast enhancement on contrast-enhanced 3D FLAIR	+	N/A	N/A	+
Cortical calcification on CT	−	N/A	+	−

ICH: intracerebral hemorrhage, TFNE: transient focal neurological episode, CAA: cerebral amyloid angiopathy, CMB: cerebral microbleed, cSS: cortical superficial siderosis, CSO: centrum semiovale, PVS: perivascular space, WMH: white matter hyperintensity, SVD: small vessel disease, FLAIR: fluid attenuated inversion recovery, N/A: not available/not performed, +: presence of the finding, −: absence of the finding.

## Data Availability

Dataset available on request from the authors.

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
