# Peer review of "Iatrogenic Cerebral Amyloid Angiopathy After Childhood Brain Surgery: Novel Findings of MRI and CT"

_2035-8377, 2025, doi:10.3390/neurolint17050064_

Round 1
Reviewer 1 Report
Comments and Suggestions for Authors
This is an interesting case series description of patients with possible iCAA.
The MRI findings are well described and the figures representatives, althoug the lack of additional critical investigations such as CSF and genetic testing as well as the lack of Amy-PET or confirmation for cadaveric dura use confirmation is, unfortunately, leaving open the possibility for other conditions. However, it must be aknowledge that the lack of these information is common for most of the reported cases in the literature, representing current situation and limitations in the field of iCAA.
I have a few suggestions and comments:
1) Page1, line 36. iCAA is "presumably" caused by a prion-like transmission.... I think that more probabilistic terms should be used for iCAA, as we are still navigating in the exploration of hypothesys for a potentially new condition. Moreover, according to currently proposed criteria for iCAA, there are additional alternative hypothesys that should merit to be cited accordingly. To this end, the manuscript with DOI 10.1136/jnnp-2022-328792 should be referenced.
2) Page 1, Lines 36, 38, 39, 41. I suggest reconsider the cited references. As above, I strongly encourage to replace ref#1 and 2 with more appropiate ones, such as: Banerjee G, Samra K, Adams ME, et al. Iatrogenic cerebral amyloid angiopathy: an emerging clinical phenomenon. Journal of Neurology, Neurosurgery Psychiatry. 2022:jnnp-2022-328792. doi:10.1136/jnnp-2022-328792
3) Page 2, Line 47. As above, the appropiate reference for currently proposed criteria is "Banerjee G, Samra K, Adams ME, et al. Iatrogenic cerebral amyloid angiopathy: an emerging clinical phenomenon. Journal of Neurology, Neurosurgery Psychiatry. 2022:jnnp-2022-328792. doi:10.1136/jnnp-2022-328792". Please replace ref#6 with the more appropiate one.
Line 45. I would like to suggest starting with the introduction of current iCAA criteria and then continue with ICH, CMBs, and cSS, concluding with the statement that these MRI markers are the same used for sporadic CAA diagnosis described in Boston 2.0 criteria.
4) Page 2, Line 65. I think that sulcal hyperintensity on contrast-enhanced 3D FLAIR images is the most interesting novelty of these cases.
I suggest adding a sentence commenting on the striking similarities with the recently described natural history of CAA-ri, where similar sulcal hyperintensities have been described and reported, including commonalities with ARIA (see Antolini et al. Neurology 2021 manuscript with DOI 10.1212/WNL.0000000000012778).
5) Page 4, Line 120. As above, I suggest adding commonalities with CAA-ri.
6) Page 6, Line 148. Please replace ref #8 and 9 with the more appropiate ref with DOI 10.1212/WNL.0000000000012778
7) Page 6, Line 160 and 161. Please replace ref #1 and #11 with the more appropiate references with DOI 10.1212/WNL.0000000000012778 and DOI 10.1212/WNL.0000000000200892
8) Page 7, Line 204. Please mention the lack of genetic testing within the limitations
9) Page 7, Conclusions. I would suggest add a comment also on the timely need for fluid biomarkers and pathology data to properly confirm or not the iCAA hypothesys and diagnosis.
Author Response
Reviewer1
The MRI findings are well described and the figures representatives, although the lack of additional critical investigations such as CSF and genetic testing as well as the lack of Amy-PET or confirmation for cadaveric dura use confirmation is, unfortunately, leaving open the possibility for other conditions. However, it must be aknowledge that the lack of these information is common for most of the reported cases in the literature, representing current situation and limitations in the field of iCAA.
I have a few suggestions and comments:
1) Page1, line 36. iCAA is "presumably" caused by a prion-like transmission.... I think that more probabilistic terms should be used for iCAA, as we are still navigating in the exploration of hypotheses for a potentially new condition. Moreover, according to currently proposed criteria for iCAA, there are additional alternative hypotheses that should merit to be cited accordingly. To this end, the manuscript with DOI 10.1136/jnnp-2022-328792 should be referenced.
Thank you for your valuable comment.
We revised the sentences in line 36-38 and add the reference as follows:
“iCAA is presumably caused by a prion-like transmission form of amyloid beta (Aβ) and occurs primarily in patients who underwent neurosurgery during childhood [1].”
“In addition, other reported causes of iCAA include the use of pituitary-derived hormones and blood transfusions [1].”
Reference [1]
Banerjee, G., K. Samra, M. E. Adams, Z. Jaunmuktane, A. R. Parry-Jones, J. Grieve, A. K. Toma, S. F. Farmer, R. Sylvester, H. Houlden, P. Rudge, S. Mead, S. Brandner, J. M. Schott, J. Collinge, and D. J. Werring. "Iatrogenic cerebral amyloid angiopathy: an emerging clinical phenomenon." J Neurol Neurosurg Psychiatry (2022): jnnp-2022-328792. https://doi.org/10.1136/jnnp-2022-328792.
2) Page 1, Lines 36, 38, 39, 41. I suggest reconsider the cited references. As above, I strongly encourage to replace ref#1 and 2 with more appropriate ones, such as: Banerjee G, Samra K, Adams ME, et al. Iatrogenic cerebral amyloid angiopathy: an emerging clinical phenomenon. Journal of Neurology, Neurosurgery Psychiatry. 2022:jnnp-2022-328792. doi:10.1136/jnnp-2022-328792
Thank you for your comment.
The original references1 and 2 have been replaced with the suggested citation.
Reference [1]
Banerjee, G., K. Samra, M. E. Adams, Z. Jaunmuktane, A. R. Parry-Jones, J. Grieve, A. K. Toma, S. F. Farmer, R. Sylvester, H. Houlden, P. Rudge, S. Mead, S. Brandner, J. M. Schott, J. Collinge, and D. J. Werring. "Iatrogenic cerebral amyloid angiopathy: an emerging clinical phenomenon." J Neurol Neurosurg Psychiatry (2022): jnnp-2022-328792. https://doi.org/10.1136/jnnp-2022-328792.
3) Page 2, Line 47. As above, the appropriate reference for currently proposed criteria is "Banerjee G, Samra K, Adams ME, et al. Iatrogenic cerebral amyloid angiopathy: an emerging clinical phenomenon. Journal of Neurology, Neurosurgery Psychiatry. 2022:jnnp-2022-328792. doi:10.1136/jnnp-2022-328792". Please replace ref#6 with the more appropriate one.
Thank you for your comment.
The original reference 6 has been replaced with the appropriate citation, which is now listed as reference 1.
Reference [1]
Banerjee, G., K. Samra, M. E. Adams, Z. Jaunmuktane, A. R. Parry-Jones, J. Grieve, A. K. Toma, S. F. Farmer, R. Sylvester, H. Houlden, P. Rudge, S. Mead, S. Brandner, J. M. Schott, J. Collinge, and D. J. Werring. "Iatrogenic cerebral amyloid angiopathy: an emerging clinical phenomenon." J Neurol Neurosurg Psychiatry (2022): jnnp-2022-328792. https://doi.org/10.1136/jnnp-2022-328792.
Line 45. I would like to suggest starting with the introduction of current iCAA criteria and then continue with ICH, CMBs, and cSS, concluding with the statement that these MRI markers are the same used for sporadic CAA diagnosis described in Boston 2.0 criteria.
Thank you for your valuable suggestion.
We revised this paragraph as follows.
“The diagnostic criteria for iCAA have been proposed [1], and they include imaging features such as lobar ICH, lobar cerebral microbleeds (CMB), and cortical superficial siderosis (cSS), which are the same as those used for the diagnosis of sporadic CAA in Boston criteria 2.0, except for non-hemorrhagic lesions [4].”
Reference [1]
Banerjee, G., K. Samra, M. E. Adams, Z. Jaunmuktane, A. R. Parry-Jones, J. Grieve, A. K. Toma, S. F. Farmer, R. Sylvester, H. Houlden, P. Rudge, S. Mead, S. Brandner, J. M. Schott, J. Collinge, and D. J. Werring. "Iatrogenic cerebral amyloid angiopathy: an emerging clinical phenomenon." J Neurol Neurosurg Psychiatry (2022): jnnp-2022-328792. https://doi.org/10.1136/jnnp-2022-328792.
Reference [4]
Charidimou, A., G. Boulouis, M. P. Frosch, J. C. Baron, M. Pasi, J. F. Albucher, G. Banerjee, C. Barbato, F. Bonneville, S. Brandner, L. Calviere, F. Caparros, B. Casolla, C. Cordonnier, M. B. Delisle, V. Deramecourt, M. Dichgans, E. Gokcal, J. Herms, M. Hernandez-Guillamon, H. R. Jäger, Z. Jaunmuktane, J. Linn, S. Martinez-Ramirez, E. Martínez-Sáez, C. Mawrin, J. Montaner, S. Moulin, J. M. Olivot, F. Piazza, L. Puy, N. Raposo, M. A. Rodrigues, S. Roeber, J. R. Romero, N. Samarasekera, J. A. Schneider, S. Schreiber, F. Schreiber, C. Schwall, C. Smith, L. Szalardy, P. Varlet, A. Viguier, J. M. Wardlaw, A. Warren, F. A. Wollenweber, M. Zedde, M. A. van Buchem, M. E. Gurol, A. Viswanathan, R. Al-Shahi Salman, E. E. Smith, D. J. Werring, and S. M. Greenberg. "The Boston Criteria Version 2.0 for Cerebral Amyloid Angiopathy: A Multicentre, Retrospective, Mri-Neuropathology Diagnostic Accuracy Study." Lancet Neurol 21, no. 8 (2022): 714-25.
4) Page 2, Line 65. I think that sulcal hyperintensity on contrast-enhanced 3D FLAIR images is the most interesting novelty of these cases.
I suggest adding a sentence commenting on the striking similarities with the recently described natural history of CAA-ri, where similar sulcal hyperintensities have been described and reported, including commonalities with ARIA (see Antolini et al. Neurology 2021 manuscript with DOI 10.1212/WNL.0000000000012778).
Thank you for your valuable comment.
We have added the sentence in Discussion as follows.
“CAA-related inflammation is reportedly part of inflammatory spectrum associated with Aβ, occasionally referred to as spontaneous amyloid-related imaging abnormalities (ARIA)-like presentations [7]. A previous study demonstrated that patients with CAA-related inflammation may show contrast enhancement on T1 post-gadolinium sequences, suggestive of leptomeningeal involvement, which closely resembles to our finding in iCAA [7]. Both iCAA and CAA related inflammation may share a common pathophysiological mechanism involving the clearance Aβ peptide [5].
Reference [7]
Antolini, L., DiFrancesco, J. C., Zedde, M., Basso, G., Arighi, A., Shima, A., Cagnin, A., Caulo, M., Carare, R. O., Charidimou, A., Cirillo, M., Di Lazzaro, V., Ferrarese, C., Giossi, A., Inzitari, D., Marcon, M., Marconi, R., Ihara, M., Nitrini, R., Orlandi, B., Padovani, A., Pascarella, R., Perini, F., Perini, G., Sessa, M., Scarpini, E., Tagliavini, F., Valenti, R., Vázquez-Costa, J. F., Villarejo-Galende, A., Hagiwara, Y., Ziliotto, N., & Piazza, F. (2021). Spontaneous ARIA-like events in cerebral amyloid angiopathy-related inflammation: A multicenter prospective longitudinal cohort study. Neurology, 97(18), e1809–e1822. https://doi.org/10.1212/WNL.0000000000012778
Reference [5]
Fandler-Hofler, S., K. Kaushik, B. Storti, S. Pikija, D. Mallon, G. Ambler, P. T. Damavandi, L. Panteleienko, I. Canavero, M. A. A. van Walderveen, E. S. van Etten, J. C. DiFrancesco, C. Enzinger, T. Gattringer, A. Bersano, M. J. H. Wermer, G. Banerjee, and D. J. Werring. "Clinical-Radiological Presentation and Natural History of Iatrogenic Cerebral Amyloid Angiopathy." J Neurol Neurosurg Psychiatry (2025): jnnp-2024-335164. doi: 10.1136/jnnp-2024-335164. Online ahead of print.
5) Page 4, Line 120. As above, I suggest adding commonalities with CAA-ri.
Thank you for your valuable comment.
This section you referring to is the case description; therefore, it may be somewhat difficult to add a sentence about commonalities with CAA related inflammation.
6) Page 6, Line 148. Please replace ref #8 and 9 with the more appropriate ref with DOI 10.1212/WNL.0000000000012778
Thank you for your comment.
We have replaced reference 8 and 9 in line 148 with your suggestion.
7) Page 6, Line 160 and 161. Please replace ref #1 and #11 with the more appropriate references with DOI 10.1212/WNL.0000000000012778 and DOI 10.1212/WNL.0000000000200892
Thank you for your comment.
We have replaced the references with your suggestion.
8) Page 7, Line 204. Please mention the lack of genetic testing within the limitations
Thank you for your comment.
We have revised the sentence in the limitations as follows.
“All cases have no confirmation of amyloid by brain biopsy, amyloid PET, or CSF ex-amination, and no genetic testing was performed.”
9) Page 7, Conclusions. I would suggest add a comment also on the timely need for fluid biomarkers and pathology data to properly confirm or not the iCAA hypotheses and diagnosis.
Thank you for your insightful comment.
We have added the sentence in Conclusion, as follows.
“Additionally, the fluid biomarkers and pathological data are needed to confirm the iCAA hypothesis and diagnosis.”
Reviewer 2 Report
Comments and Suggestions for Authors
Great cases. I have several points (in order of appearance in the manuscript) that could improve the manuscript.
Abstract and Discussion: Fandler-Höfler et al. report inflammation as an incidental finding. It is not clear why contrast enhancement should be a marker of advanced iCAA.
Introduction: Line 40- but can also occur later (Pantelenko et al.2024) Please discuss.
Line 45: But also non-hemorrhagic markers.
Case report: Line 45- episodes of confusion
Case 3: Was genetic testing performed? If not, an alternative possibility would be hereditary CAA with affiliated findings.
Are the calcifications visible on T2*?
Figure 1b: Is that a cerebellar parenchymal defect?
Table 1: The authors should distinguish between not available/not performed (N/A) and not present.
The authors could check for cerebellar cSS (as described in doi: 10.3238/arztebl.m2023.0215) as a sign for advanced iCAA and cerebellar CMB.
The authors should also comment on a possible lateralization of the imaging markers with regards to the site of surgery.
The authors should shortly explain the SVD-score and give a range for the non-expert reader.
Author Response
Great cases. I have several points (in order of appearance in the manuscript) that could improve the manuscript.
Abstract and Discussion: Fandler-Höfler et al. report inflammation as an incidental finding. It is not clear why contrast enhancement should be a marker of advanced iCAA.
Thank you for your valuable comment.
In the report by Fandler-Höfler et al., neuroimaging findings suggesting inflammation include cortical oedema, parenchymal and sulcal hyperintensities. In addition, the case of Figure 5 in their article showed leptomeningeal enhancement on contrast-enhanced T1-weighted image. We agree that these findings are associated with inflammation.
On the other hand, our case 1 and 4 showed no other findings, such as cortical edema, parenchymal and sulcal hyperintensities, but sulcal contrast enhancement on contrast-enhanced FLAIR. Therefore, we consider another cause of sulcal enhancement, namely, increased meningeal vascular permeability due to disruption of the blood-meningeal barrier, leading to local leakage of gadolinium-based contrast agents into the sub-arachnoid space, as noted in Discussion section. We have considered advanced iCAA as high SVD score, regardless of symptoms. Our case 1 and 4 have high SVD scores, which indicates advanced iCAA.
Introduction: Line 40- but can also occur later (Pantelenko et al.2024) Please discuss.
Thank you for your comment.
We have revised the sentence as follows, and added the recommended reference.
“The onset of iCAA tends to occur before the age of 55, but can occur at the age 65 or older, depending on the timing of surgery and a latency period of 20–40 years [1,2].”
Reference [1]
Banerjee, G., K. Samra, M. E. Adams, Z. Jaunmuktane, A. R. Parry-Jones, J. Grieve, A. K. Toma, S. F. Farmer, R. Sylvester, H. Houlden, P. Rudge, S. Mead, S. Brandner, J. M. Schott, J. Collinge, and D. J. Werring. "Iatrogenic cerebral amyloid angiopathy: an emerging clinical phenomenon." J Neurol Neurosurg Psychiatry (2022): jnnp-2022-328792. https://doi.org/10.1136/jnnp-2022-328792.
Reference [2]
Kaushik, K., E. S. van Etten, B. Siegerink, L. J. Kappelle, A. W. Lemstra, Fhbm Schreuder, C. J. M. Klijn, W. C. Peul, G. M. Terwindt, M. A. A. van Walderveen, and M. J. H. Wermer. "Iatrogenic Cerebral Amyloid Angiopathy Post Neurosurgery: Frequency, Clinical Profile, Radiological Features, and Outcome." Stroke 54, no. 5 (2023): 1214-23.
Line 45: But also non-hemorrhagic markers.
Thank you for your comment.
We have revised the sentence and incorporated the suggestion from both Reviewer 1 and 2 as follows.
“The diagnostic criteria for iCAA have been proposed [1], and they include imaging features such as lobar ICH, lobar cerebral microbleeds (CMB), and cortical superficial siderosis (cSS), which are the same as those used for the diagnosis of sporadic CAA in Boston criteria 2.0, except for non-hemorrhagic lesions [4].”
Reference [1]
Banerjee, G., K. Samra, M. E. Adams, Z. Jaunmuktane, A. R. Parry-Jones, J. Grieve, A. K. Toma, S. F. Farmer, R. Sylvester, H. Houlden, P. Rudge, S. Mead, S. Brandner, J. M. Schott, J. Collinge, and D. J. Werring. "Iatrogenic cerebral amyloid angiopathy: an emerging clinical phenomenon." J Neurol Neurosurg Psychiatry (2022): jnnp-2022-328792. https://doi.org/10.1136/jnnp-2022-328792.
Reference [4]
Charidimou, A., G. Boulouis, M. P. Frosch, J. C. Baron, M. Pasi, J. F. Albucher, G. Banerjee, C. Barbato, F. Bonneville, S. Brandner, L. Calviere, F. Caparros, B. Casolla, C. Cordonnier, M. B. Delisle, V. Deramecourt, M. Dichgans, E. Gokcal, J. Herms, M. Hernandez-Guillamon, H. R. Jäger, Z. Jaunmuktane, J. Linn, S. Martinez-Ramirez, E. Martínez-Sáez, C. Mawrin, J. Montaner, S. Moulin, J. M. Olivot, F. Piazza, L. Puy, N. Raposo, M. A. Rodrigues, S. Roeber, J. R. Romero, N. Samarasekera, J. A. Schneider, S. Schreiber, F. Schreiber, C. Schwall, C. Smith, L. Szalardy, P. Varlet, A. Viguier, J. M. Wardlaw, A. Warren, F. A. Wollenweber, M. Zedde, M. A. van Buchem, M. E. Gurol, A. Viswanathan, R. Al-Shahi Salman, E. E. Smith, D. J. Werring, and S. M. Greenberg. "The Boston Criteria Version 2.0 for Cerebral Amyloid Angiopathy: A Multicentre, Retrospective, Mri-Neuropathology Diagnostic Accuracy Study." Lancet Neurol 21, no. 8 (2022): 714-25.
Case report: Line 45- episodes of confusion
Thank you for your comment.
Unfortunately, we are not sure how to revise this section.
Case 3: Was genetic testing performed? If not, an alternative possibility would be hereditary CAA with affiliated findings.
Thank you for your valuable comment.
All our cases did not perform genetic testing.
Thus, we revised the sentence in the limitation paragraph, as Reviewer 1 and 2 suggested.
“All cases have no confirmation of amyloid by brain biopsy, amyloid PET, or CSF ex-amination, and no genetic testing was performed.”
Are the calcifications visible on T2*?
Thank you for pointing out.
We have revised the sentence in the description of Case 3 as follows.
“CT showed prominent cortical calcifications in both occipital lobes, which were not visible on T2*-weighted image.”
Figure 1b: Is that a cerebellar parenchymal defect?
Thank you for your comment on the Figure.
We confirmed a cerebellar parenchymal defect in this case, although detailed information regarding the surgical procedure is not available due to the lack of clinical documentation maintained within the hospital.
We have revised the sentence in the description of Case 3 description, as follows.
“Use of cadaveric dura mater, Lyodura, was confirmed, although any other detailed in-formation about surgical procedure was not available.”
Table 1: The authors should distinguish between not available/not performed (N/A) and not present.
Thank you for your comment.
We revised the phrase “N/A, not available” to “N/A, not available/not performed”, and added “+ presence of the finding, - absence of the finding” in the Table footnote.
The authors could check for cerebellar cSS (as described in doi: 10.3238/arztebl.m2023.0215) as a sign for advanced iCAA and cerebellar CMB.
Thank you for your valuable comment.
We revised the sentence in the description of Case 3 as follows.
“T2*-weighted images showed no cSS in the right precentral gyrus or other gyri, while multiple lobar CMBs were detected mainly in both cerebral hemispheres, with several CMBs also observed in the cerebellum.”
Additionally, we revised the sentence in line 200-202 in Discussion section as follows.
“Case 1 and 3 showed several cerebellar CMBs. Cerebellar cSS and CMBs are reportedly signs for advanced iCAA [15].”
Reference [15]
Jensen-Kondering, U., Heß, K., Flüh, C., Kuhlenbäumer, G., & Margraf, N. G. (2024). A rare case of iatrogenic prion-like pathogenesis of cerebral amyloid angiopathy. Dtsch Arztebl Int, 121(2), 68–69. https://doi.org/10.3238/arztebl.m2023.0215
The authors should also comment on a possible lateralization of the imaging markers with regards to the site of surgery.
Thank you for your valuable comment.
We have added the sentence in line 202-204 in Discussion section as follows.
“CMBs and cSS in iCAA tend to occur near the site of surgery, primarily extending to both cerebral hemispheres and occasionally to the cerebellum.”
The authors should shortly explain the SVD-score and give a range for the non-expert reader.
Thank you for your valuable comment.
We have added the explanation of the SVD score in line 49-54 in Introduction as follows.
“As CAA is considered a type of small vessel disease (SVD), previous literature proposed the CAA-SVD scoring system, which consists of lobar CMB, cSS, perivascular spaces in the centrum semiovale (CSO-PVS), and white matter hyperintensity (WMH) [6]. This score ranges from 0 to 6, with higher scores indicating greater severity of SVD, which is useful for sporadic CAA [6].”
Round 2
Reviewer 2 Report
Comments and Suggestions for Authors
Case report: Line 45- episodes of confusion
Thank you for your comment.
Unfortunately, we are not sure how to revise this section.
What I mean is to describe the symptom as a syndrome, i.e. "episodes of confusion" or "episodes of loss of orientation".
Additionally, we revised the sentence in line 200-202 in Discussion section as follows.
“Case 1 and 3 showed several cerebellar CMBs. Cerebellar cSS and CMBs are reportedly signs for advanced iCAA [15].”
...should be "Cerebellar cSS and CMBs are reportedly signs for advanced iCAA [15]."
Author Response
Case report: Line 45- episodes of confusion
Thank you for your comment.
Unfortunately, we are not sure how to revise this section.
What I mean is to describe the symptom as a syndrome, i.e. "episodes of confusion" or "episodes of loss of orientation".
Thank you for articulating your question.
We have revised the Case 1 description as follows.
“A 47-year-old woman presented with severe cognitive impairment, including episodes of not knowing where her house was, indicative of loss of orientation.”
Additionally, we revised the sentence in line 200-202 in Discussion section as follows.
“Case 1 and 3 showed several cerebellar CMBs. Cerebellar cSS and CMBs are reportedly signs for advanced iCAA [15].”
...should be "Cerebellar cSS and CMBs are reportedly signs for advanced iCAA [15]."
Thank you for your comment.
We have omitted the words “and CMBs” as follows.
“Cerebellar cSS is reportedly a sign for advanced iCAA”